# Insights into the mechanism of phospholipid hydrolysis by plant non-specific phospholipase C

Ruyi Fan[1,2,6], Fen Zhao[1,6], Zhou Gong [3,6], Yanke Chen[3], Bao Yang[1], Chen Zhou[1], Jie Zhang[1], Zhangmeng Du[1], Xuemin Wang [4,5], Ping Yin [1], Liang Guo [1] & Zhu Liu [1,2]

Non-specific phospholipase C (NPC) hydrolyzes major membrane phospholipids to release diacylglycerol (DAG), a potent lipid-derived messenger regulating cell functions. Despite extensive studies on NPCs reveal their fundamental roles in plant growth and development, the mechanistic understanding of phospholipid-hydrolyzing by NPCs, remains largely unknown. Here we report the crystal structure of Arabidopsis NPC4 at a resolution of 2.1 Å. NPC4 is divided into a phosphoesterase domain (PD) and a C-terminal domain (CTD), and is structurally distinct from other characterized phospholipases. The previously uncharacterized CTD is indispensable for the full activity of NPC4. Mechanistically, CTD contributes NPC4 activity mainly via CTD$^{\alpha 1}$-PD interaction, which ultimately stabilizes the catalytic pocket in PD. Together with a series of structure-guided biochemical studies, our work elucidates the structural basis and provides molecular mechanism of phospholipid hydrolysis by NPC4, and adds new insights into the members of phospholipase family.

Phospholipases are diverse lipolytic enzymes that catalyze hydrolysis of specific ester bonds in phospholipids, and are abundant in all living organisms[1]. These enzymes play critical roles in cell signaling, lipid metabolism, and membrane remodeling[2,3]. According to the cleavage site of phospholipid, phospholipases are grouped into phospholipase A1 (PLA1), phospholipase A2 (PLA2), phospholipase C (PLC) and phospholipase D (PLD) (Fig. 1a)[4].

PLC hydrolyzes a phosphodiester bond at the glycerol side of a phospholipid molecule to release corresponding phosphorylated head group and diacylglycerol (DAG), a potent lipid-derived second messenger. The PLC-derived DAG regulates various cellular and physiological functions[5,6]. Two types of PLCs have been identified, the

phosphoinositide-specific PLC (PI-PLC) and the non-specific PLC (NPC) (Fig. 1a). PI-PLC is evolutionarily conserved and specifically targets phosphoinositides (PIPs) for hydrolysis[3,7]. In contrast, NPC harbors promiscuous activities, hydrolyzing major membrane phospholipids, such as glycosyl inositol phosphoryl ceramide (GIPC), phosphatidylcholine (PC), phosphatidylethanolamine (PE), and phosphatidylserine (PS)[8,9]. Unlike PI-PLC occurring in various organisms, NPC has only been found in bacteria and plants[2,10,11]. Molecular mechanisms of eukaryotic PI-PLCs have been well elucidated[12,13]. PI-PLCs contain multiple domains, including PH domain, EF-hand domain, catalytic X-Y domain and C2 domain. Specifically, the PH domain tethers PI-PLC to the membrane via specific binding to

[1]National Key Laboratory of Crop Genetic Improvement, Hubei Hongshan Laboratory, Huazhong Agricultural University, Wuhan 430070, China. [2]Shenzhen Branch, Guangdong Laboratory for Lingnan Modern Agriculture, Genome Analysis Laboratory of the Ministry of Agriculture, Agricultural Genomics Institute at Shenzhen, Chinese Academy of Agricultural Sciences, Shenzhen 518124, China. [3]Innovation Academy for Precision Measurement Science and Technology, Chinese Academy of Sciences, Wuhan 430071, China. [4]Department of Biology, University of Missouri, St. Louis, MO 63121, USA. [5]Donald Danforth Plant Science Center, St. Louis, MO 63132, USA. [6]These authors contributed equally: Ruyi Fan, Fen Zhao, Zhou Gong. ✉e-mail: guoliang@mail.hzau.edu.cn; liuzhu@mail.hzau.edu.cn

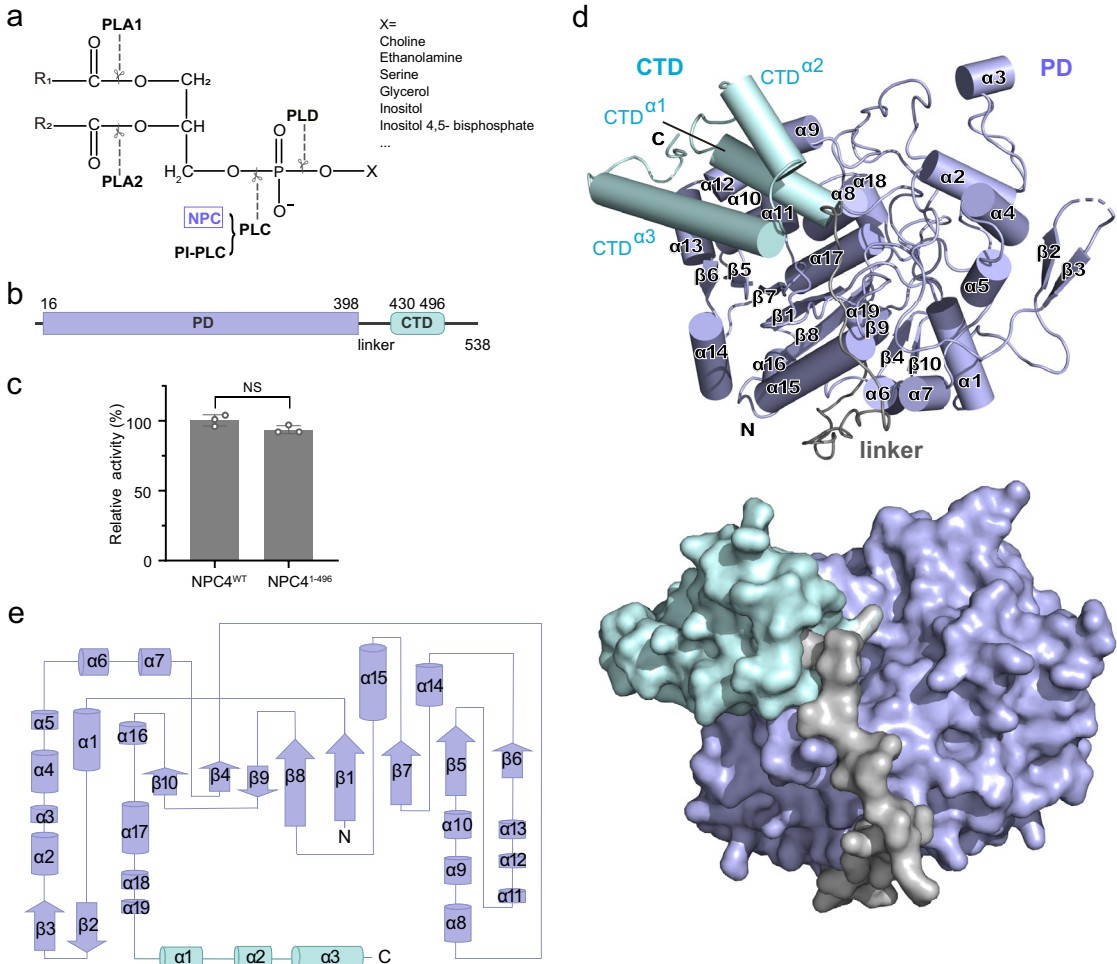

**Fig. 1 | Overall structure of Arabidopsis NPC4. a** Phospholipid structure and hydrolysis site by phospholipases. PLC is divided into NPC and PI-PLC, based on their substrate preferences. R1 and R2 represent two fatty acyl chains. X represents the head group of phospholipid. **b** Schematic depiction of NPC4. Phosphoesterase domain (PD, residues 16–398) and C-terminal domain (CTD, residues 430–496) are connected by a linker (residues 399–429). **c** Activity assay. Relative enzyme activity of NPC4[1–496] is referenced to NPC4[WT]. NPC4[WT] is the full-length NPC4. NPC4[1–496] (containing residues 1–496) is a border that is used for crystallization. The data from three independent measurements are averaged, and the error indicates SD. Statistical analysis: unpaired two-tailed *t*-test. NS, not significant. Source data are provided as a Source Data file. **d** Structure of NPC4[1–496]. PD, linker and CTD are colored in lightblue, gray and palecyan, respectively. The α-helices and β-strands are numbered and highlighted. The cartoon (up) and surface (down) representations are displayed in the same perspective. **e** Secondary structure and topological diagrams of NPC4[1–496], with labeled structure elements.

phosphatidylinositol 4,5-bisphosphate (PIP$_2$), and the calcium-dependent lipid binding C2 domain fixes the catalytic domain in a productive orientation relative to the membrane for substrate entry and hydrolysis. Calcium is necessary for both the EF-hand facilitated lipid – C2 binding and for the catalysis reaction.

In contrast, NPCs encompass only a conserved phosphoesterase domain (named PD here after) and an uncharacterized C-terminal domain (named CTD here after), lacking PH domain, EF-hand domain and C2 domain that occur in PI-PLCs (Fig. 1b). These differences indicate that NPCs use different structural and biochemical mechanisms for functioning. Accordingly, NPCs were reported to be calcium-independent phospholipases[11,14]. Arabidopsis genome has six *NPC*s that are involved in various plant-conserved biological processes, including lateral root growth[15] and gametophyte development[16], seed oil production[17], and plant response to various stressors[14,18–20]. In particular, NPC4 is highly induced under various stress conditions and involved in plant response to salt, abscisic acid, and phosphate limitation[8,11,21,22]. Recently, plant NPC4 has been found to hydrolyze the major phosphosphingolipid GIPC in response to phosphate deficiency[8], and that NPC4 is tethered to the plasma membrane via a *S*-acylation occurred on a cysteine (Cys533) at the C terminus[23].

Since the discovery of NPCs, they have gained a lot of interest. This is because they are a novel type of plant phospholipases and play various fundamental roles in plant growth and development. Moreover, recently papers in the area have found that NPC4 can help plant to cope with phosphate limitation[8,23]. Therefore, while the interest of NPCs are clear, we do not know how they really work. And, among various classes of phospholipases (including PLA1, PLA2, PLD, PI-PLC, and NPC), eukaryotic NPC is the only one whose structure and working mechanism have remained uncharacterized. In this work, we solve this mystery and define the molecular basis of how NPC4 works, and provide new mechanistic insights into the members of phospholipase family, by resolving the experimental structure of Arabidopsis NPC4 and performing a series of structure-guided biochemical studies.

## Results

### Structure of NPC4

To determine the structure of NPC, we screened all six NPC members of Arabidopsis. After numerous attempts, we could crystallize a border of NPC4[1–496] that presents enzyme activity similar to wild-type (WT) NPC4 (Fig. 1b, c, Supplementary Fig. 1 and Methods). A set of crystals were matured in about 7 days, and the structure of NPC4[1–496] was

determined by single-wavelength anomalous dispersion method using a selenomethionine derived protein crystal and refined to 2.1 Å resolution (Fig. 1d, Supplementary Table 1). The overall structure can be divided into two domains, the PD (residues 16–398) and the CTD (residues 430–496) (Fig. 1d). The PD assembles 19 α-helices and 10 β-strands, and mainly adopts a sandwiched topology (Fig. 1d, e). A β-sheet made of β1 and β4-β10 strands is sandwiched between α1, 6, 7, 14–16 helices and α8-α13, α17, α18 helices. The other β-sheet (paired β2, β3) and α2-α5 are alongside the sandwich (Fig. 1e). The CTD encompasses 3 α-helices (CTD$^{\alpha 1}$, CTD$^{\alpha 2}$ and CTD$^{\alpha 3}$) and associates with PD, connected by a hugged linker (residues 399–429) (Fig. 1d).

A 3-dimensional structural homology search with the program DALI[24] revealed that the structure of NPC4 has not been previously observed with other phospholipases, supporting the notion that NPC4 is a novel phospholipase different from other phospholipases[11]. Instead, the closest structural homology is with the acid phosphatase A (AcpA) from *Francisella tularensis*[25], an enzyme hydrolyzing phosphate monoester (Supplementary Fig. 2). Although NPC4 has low sequence identity with AcpA and harbors the additional CTD domain that doesn't occur in AcpA (Supplementary Fig. 3), the PD domain of NPC4 folds similar to AcpA with a root mean square deviation (RMSD) of 3.27 Å, as well as the key residues for catalysis in the catalytic pockets (Supplementary Fig. 4). As these similar phosphoesterase domain structures take different substrate preferences, we infer that NPC4 would use a different mechanism for targets recognition and/or hydrolysis, wherein the extra CTD domain in NPC4 is indispensible (see below).

## Catalytic mechanism of phospholipid-hydrolyzing by NPC4

NPC4 shapes a negatively charged cleft (Fig. 2a), where previously predicted key residues (E23, N24, H79, T158, H264, D299 and E300) responsible for catalysis[2] are distributed, at the interface of the β1 strand, α2 and α7 helices, β9 and β10 strands, with successive E23 and N24 locate at the end of β1 strand, H79 locate at the tip of α2 helix, T158 locate at the tip of α7 helix, H264 locate at the tip of β7, and successive D299 and E300 located at the end of β8 strand (Supplementary Fig. 5). Alanine substitution of any above-mentioned residues in the catalytic pocket dramatically reduced or abolished the activity of NPC4 (including E23A, N24A, H79A, T158A, H264A, D299A and E300A) (Fig. 2b, Supplementary Fig. 1), indicating their crucial roles in substrate hydrolysis. A funneled-negative charged cleft leading to the catalytic pocket was also found in other eukaryotic phospholipase members including phospholipase D[26–28], PI-PLC[12], and phospholipase A2[29], indicating that it is a common feature among eukaryotic phospholipases.

Furthermore, we found that an extra electron density can be well built with a phosphothreonine replacing T158 in the catalytic pocket (Supplementary Figs. 5, 6). The position of this phosphate group on the modified T158 overlaps with an orthovanadate inhibitor observed in AcpA that blocks nucleophilic attack on substrate[25] (Supplementary Fig. 4), suggesting that T158 acts as the nucleophile in NPC4. Similar site-specific phosphorylation of an alternative nucleophile, a histidine, in catalytic pocket was also observed in PLD[26,30]. In PLD this phosphohistidine was identified to present a transition state during the hydrolysis of phospholipid substrates, that is formed after the first nucleophilic attack of a substrate by the catalytic histidine[26,30–33]. And subsequently, the formed phosphoenzyme intermediate is further attacked by an activated water to release the product[26,30–33]. Without a substrate, the phosphohistidine was suggested to reflect an autoinhibited state of PLD[26], as the phosphorylation blocks the nucleophilic atom to initiate the first attack. Based on our structure observation, in the case of NPC4 it should be the oxygen side-chain atom of T158 that performs the first nucleophilic attack, and its phosphorylation would block the substrate-hydrolysis activity of NPC4. Supporting

this, a phospho-mimetic mutation of T158 (T158E or T158D) eliminated NPC4 activity (Fig. 2b, Supplementary Fig. 1).

Outside the catalytic pocket of NPC4, another phosphate is observed in the structure (Supplementary Fig. 5). This is reminiscent of that this structure was crystalized in the presence of $KH_2PO_4$ (Methods). To assess whether phosphate affects NPC4 activity, we checked this by adding $KH_2PO_4$ into the buffer of activity assay. We found that the activity of NPC4 changed little by adding $KH_2PO_4$ (Supplementary Figs. 1, 7). Thus, the observed phosphate most likely aids NPC4 crystallization, but is not required for NPC4 activity.

There is a metal ion bound in the catalytic pocket of NPC4 crystal structure, based on the observation of a clear divalent cation electron density (Supplementary Fig. 8a). Residues E23, N24, phosphorylated-T158, D299 and E300 bind with this metal ion via five coordinates (Supplementary Fig. 8b). Given that we observe this metal ion in the crystal structure, previous studies found that metal ions were not required for NPCs' activity[11,17,21]. To clarify if this co-crystallized metal ion is involved in substrate hydrolysis, we prepared NPC4 in the presence of ethylenediaminetetraacetic acid (EDTA) or divalent metal ions, and assessed their effect on the enzyme activity (Supplementary Figs. 1, 8c). By adding 2 mM EDTA into the activity assay buffer or maintaining 2 mM EDTA during the protein purification, for chelating potential divalent metal ions bound in protein, we observed no changes in NPC4 activity compared to that in the absence of EDTA (Supplementary Fig. 8c). Furthermore, increasing the added EDTA to a higher concentration of 5 mM still did not reduce the enzyme activity (Supplementary Fig. 8c), indicating no involvement of metal ions in catalysis. In parallel, we speculated that if a metal ion is involved in catalysis, adding ions to the assay buffer would increase NPC4 activity. By adding 1 mM divalent metal ions, such as $Ca^{2+}$, $Mg^{2+}$ or $Zn^{2+}$, into the activity assay buffer, we found that the enzyme activity did not increase (Supplementary Fig. 8c), suggesting that these ions are not required for NPC4 catalysis. The presence of $Zn^{2+}$ partially reduced the enzyme activity (Supplementary Fig. 8c), possibly due to protein instability caused by $Zn^{2+}$ or other unknown reasons. Inhibition of $Zn^{2+}$ on NPC4 activity was also found in previous reports[11]. Together, it is most likely that the observed metal ion in our crystal structure is unnecessary for NPC4 activity, consistent with previous findings that the NPC4 activity was ion independent[11,21].

Collectively, a catalytic mechanism of phospholipid-hydrolyzing by NPC4 can be proposed based on our structural and mutational analyses (Fig. 2c). The oxygen side-chain atom of T158 is activated by E23, D299 and E300 (serve as general bases) and acts the first nucleophile that attacks the phosphorus atom of the substrate, while the H79 acts as the acid protonating the oxygen atom of the leaving diacylglycerol moiety. This results in the formation of phosphoenzyme intermediate with a covalent P-O bond to T158. Subsequently, an activated water, deprotonated by the H264, acts the second nucleophile that attacks the phosphoenzyme intermediate to release the product. The D76 hydrogen-bound to the two histidines (H79 and H264) in the structure (Fig. 2a), functioning to stabilize them in different ionic forms for catalysis. Consistent with this, alanine substitution of D76 (D76A) reduced NPC4 activity (Fig. 2b, Supplementary Fig. 1). The N24 likely stabilizes the catalytic pocket and the intermediate, by forming a network of hydrogen bonds (Fig. 2a). Above-mentioned residues are highly conserved in plant NPCs (Supplementary Fig. 3), and any single point mutation of them impacts NPC4 activity (Fig. 2b), indicating their crucial roles for function.

## CTD contributes NPC4 activity via CTD$^{\alpha 1}$-PD interaction

During NPC4$^{1–496}$ crystallization, another set of crystals were obtained after about 2 months (Methods). A structure of this crystal was determined at 2.1 Å resolution (Supplementary Fig. 9a, Supplementary Table 1). Residues encompassing PD (residues 12–258 and 264–415)

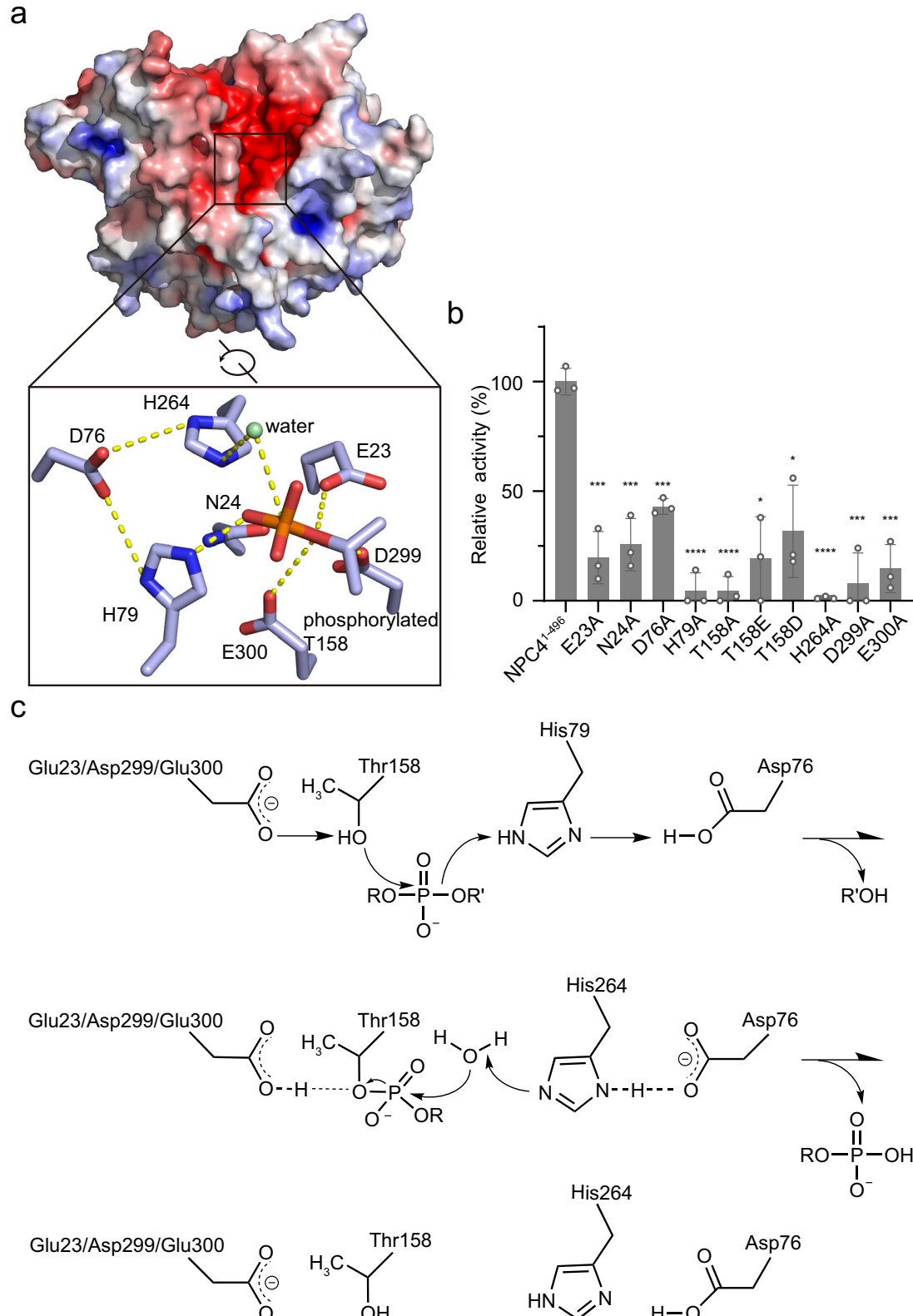

can be well built in the crystal (NPC4[1–415]), whereas no electron densities (including residues 416–496) are observed for CTD region (Supplementary Fig. 9a, b). Unlike the structure of NPC[1–496], no phosphorylation, metal ion and free phosphate molecule occur in this structure. By collecting crystals of the determined NPC4[1–415] structure, SDS-PAGE results shown that the protein was partially degraded during crystallization (Supplementary Fig. 10), possibly resulting in the omitted CTD in the crystal. Comparing the crystallizing conditions of NPC4[1–496] and NPC4[1–415] structures (Methods), KH$_2$PO$_4$ was absent in the NPC4[1–415] crystallization. We thus suspected that this salt might enhance NPC4 stability. Supporting this hypothesis, different scanning fluorimetry (DSF) analysis and limited proteolysis experiments showed that NPC4 was more stable in the presence of KH$_2$PO$_4$ (Supplementary Fig. 11).

**Fig. 2 | The active site and catalytic mechanism of NPC4. a** Electrostatic surface of NPC4$^{1-496}$ is colored in terms of electrostatic potential, displayed in a scale from red (−5 kT/e) to blue (+5 kT/e). Key residues of NPC4 involved in substrate hydrolysis and their interaction network are zoomed-in and shown in stick representation. The T158-linked phosphate is shown in stick representation, and the water responsible for the second nucleophilic attack is shown as a green sphere. **b** Activity assay. Relative enzyme activity of each point mutant is referenced to NPC4$^{1-496}$. The data from three independent measurements are

averaged, and the error indicates SD. Statistical analysis: unpaired two-tailed *t*-test. Source data are provided as a Source Data file. $^*p < 0.05$, $^{***}p < 0.001$, $^{****}p < 0.0001$. $p$ (E23A) = 0.0005, $p$ (N24A) = 0.0007, $p$ (D76A) = 0.0002, $p$ (H79A) < 0.0001, $p$ (T158A) < 0.0001, $p$ (T158E) = 0.0022, $p$ (T158D) = 0.0058, $p$ (H264A) < 0.0001, $p$ (D299A) = 0.0005 and $p$ (E300A) = 0.0003. **c** Schematic representation for the proposed chemical mechanism of phospholipid-hydrolyzing by NPC4. R represents the head group of a phospholipid and R′ represents the diacylglycerol moiety.

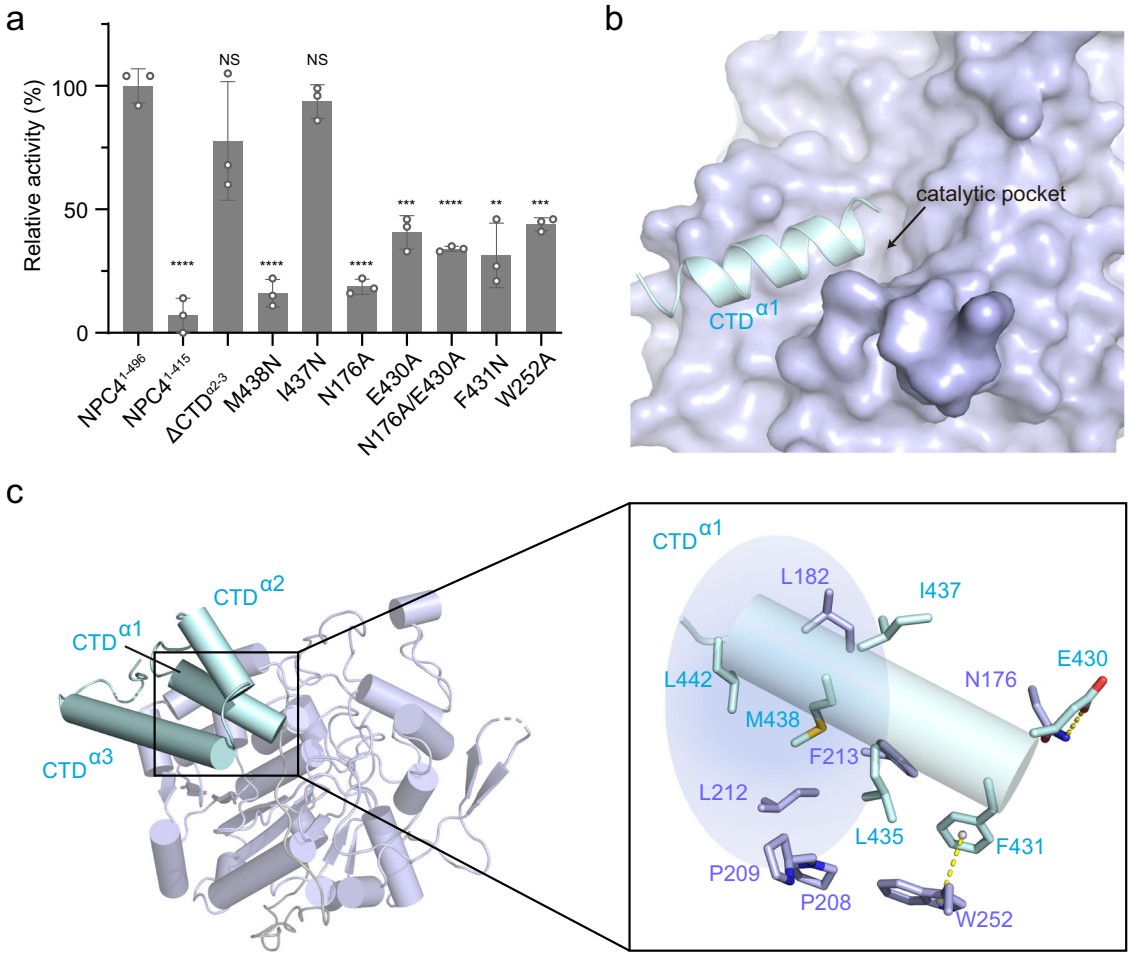

**Fig. 3 | CTD contributes NPC4 activity via CTD$^{\alpha 1}$-PD interaction. a** Activity assay. Relative enzyme activity of each construct or mutant is referenced to NPC4$^{1-496}$. The data from three independent measurements are averaged, and the error indicates SD. ΔCTD$^{\alpha 2-\alpha 3}$ contains residues 1–443 of NPC4, lacking the α-helix 2 and 3 of CTD. Statistical analysis: unpaired two-tailed *t*-test. NS, not significant. $^{**}p < 0.01$, $^{***}p < 0.001$, $^{****}p < 0.0001$. $p$ (NPC41-415) <0.0001, $p$ (ΔCTDα2-α3) = 0.1965, $p$ (M438N) < 0.0001, $p$ (I437N) = 0.2184,

$p$ (N176A) < 0.0001, $p$ (E430A) = 0.0003, $p$ (N176A/E430A) < 0.0001, $p$ (F431N) = 0.0013 and $p$ (W252A) = 0.0002. Source data are provided as a Source Data file. **b** CTD$^{\alpha 1}$ leads to the entrance of the catalytic pocket in NPC4$^{1-496}$. CTD$^{\alpha 1}$ and PD are shown in surface and cartoon representation, respectively. **c** Interaction interface between CTD$^{\alpha 1}$ and PD. The zoomed-in box illustrates the detailed interactions. Blue shading denotes the hydrophobic pocket formed between CTD$^{\alpha 1}$ and PD.

Superposing the determined structure of NPC4$^{1-415}$ to NPC4$^{1-496}$, the architectures of PD are basically the same but with a significant local conformational change in a loop (including residues 250–268) (Supplementary Fig. 9b). The catalytic residue H264 (Fig. 2c) in this loop twists out of the catalytic pocket in the NPC4$^{1-415}$ structure (with a measured distance of 8.4 Å), compared to the NPC4$^{1-496}$ structure (Supplementary Fig. 9b, c). Correspondingly, other active site residues are deflected to a certain extent (Supplementary Fig. 9c). We thus wondered whether the truncated structure (NPC4$^{1-415}$), lacking CTD, maintains enzyme activity. Then we constructed this border and measured its enzyme activity. It showed that the substrate hydrolyzing-activity of NPC4$^{1-415}$ was almost eliminated (Fig. 3a,

Supplementary Fig. 1). Although our NPC4$^{1-496}$ structure shows that CTD is located outside the catalytic pocket of PD (Fig. 1d) and previous studies have indicated that CTD was not directly involved in catalysis[2], our results suggest that CTD is required for NPC4 activity.

In the structure of NPC4$^{1-496}$, the initial fourteen residues of CTD form an α-helix (CTD$^{\alpha 1}$, including residues 430–443) leading to the entrance of the catalytic pocket (Fig. 3b). A deletion of the entire CTD eliminated the enzyme activity, whereas removal of the last two helices of CTD (ΔCTD$^{\alpha 2-3}$) impaired little (Fig. 3a, Supplementary Fig. 1). This indicates that CTD$^{\alpha 1}$ of CTD is the key element responsible for NPC4 activity. CTD$^{\alpha 1}$ associates with PD mainly via a hydrophobic interaction and a paired hydrogen bond (Fig. 3c). The hydrophobic network

encompasses L435, M438 and L442 in CTD$^{\alpha1}$, and L182, P208, P209, L212 and F213 in PD. A substitution of the centered M438 to hydrophilic asparagine (M438N) in the hydrophobic pocket significantly reduced the activity of NPC4, whereas a substitution on CTD$^{\alpha1}$ outside the hydrophobic pocket, I437N, had little impact on the activity (Fig. 3a, c). In the interaction interface, the carboxylic acid group of E430 hydrogen bonding to the side chain nitrogen atom of N176 (Fig. 3c). Substituting one or both of them with alanine (N176A, E430A or N176A/E430A) largely reduced the activity of NPC4 (Fig. 3a). Furthermore, PD-located W252 interacts with F431 in CTD$^{\alpha1}$, by forming a π-π stacking interaction (Fig. 3c). Substituting F431 to asparagine (F431N), or W252 to alanine (W252A), dramatically reduced the activity of NPC4 (Fig. 3a). These revealed activity-indispensable residues (including N176 and E430, and hydrophobic W252, F431 and M438) are highly conserved in NPCs (Supplementary Fig. 3). Taken together, our structural analyses and biochemical evidences define a crucial role of CTD for the full activity of NPC4, that is CTD may stabilize the catalytic pocket in PD via CTD$^{\alpha1}$-PD interaction.

## Modeling NPC4 targets to different substrates

Non-specific phospholipase C (NPC) harbors promiscuous activities that hydrolyzes major membrane phospholipids, such as phosphoglycerolipid GIPC and phosphosphingolipid PC[9]. To directly visualize substrates recognition by NPC4, we tried to determine the structure of NPC4 in complex with substrates. However, by co-crystallizing different phospholipids (GIPC, PC) with NPC4 or by soaking NPC4 crystals with these substrates, none of them worked. To understand how NPC4 targets various substrates, we then docked different substrates, GIPC, PC, PE and PS, into our resolved NPC4$^{1-496}$ structure using AutoDock Vina[34], respectively (Fig. 4a–d and Methods). The docked complexes show that the four substrates are targeted into the funneled-negative charged cleft leading to the catalytic pocket. Molecular dynamics (MD) simulations of these docked enzyme-substrate models revealed that these models are stable within a simulation time of 200 ns, indicated by small RMSD fluctuations of the enzyme (Fig. 4e–h) and the associated substrate (Fig. 4i–l) in docked models. Moreover, the distances between the nucleophilic atom (oxygen side-chain atom of T158) and the phosphorus atom of substrates also slightly fluctuate during the simulations (Supplementary Fig. 12a–d), further indicating that these docked models should be stable. The exposed cleft on NPC4 is larger than the molecular size of targeted substrates, and the head groups point toward the catalytic pocket in alternative positions (Fig. 4a–d). This indicates a possible mechanism for the promiscuous activity of NPC4: the large-exposed cleft of NPC4 enables the recognition of various substrates with different molecular sizes.

Based on these docked enzyme-substrate models, we analyzed whether the conformations are competent for catalysis. The oxygen side-chain atom of T158 is responsible for the first nucleophilic attack onto the phosphorus atom of a substrate (Fig. 2c). In the docked models, the distances between the nucleophilic atom and the phosphorus atom of GIPC, PC, PE and PS substrates are 12.0 Å, 8.5 Å, 7.2 Å and 8.5 Å, respectively (Supplementary Fig. 12e–h), where the nucleophilic attack cannot happen. We thus propose that the docked conformation of a substrate should represent an initial-recognized state, and the substrate is about to be pulled into the catalytic pocket for cleavage, orchestrated by some enzyme conformational changes.

## Discussion

Among various classes of phospholipases, including PLA1, PLA2, PLD, PI-PLC, and NPC, eukaryotic NPC is the only one whose structure and working mechanism have remained uncharacterized. Here we report an experimental structure in the class of eukaryotic NPC, the Arabidopsis NPC4. NPC4 adopts a unique conformation distinct from other characterized phospholipase members[12,26,27,29]. Together with a series of structure-guided biochemical analyses, this work helps us better understand the function of NPCs and provides new mechanistic insights into the phospholipase family.

We demonstrate the active site of NPC4 and define a two-step catalytic mechanism for the substrate hydrolysis (Fig. 2a, c). This two-step mechanism is similar to that observed in PLDs[26–28,30,33], wherein a catalytic residue is deprotonated and activated to perform the first nucleophilic attack onto the phosphorus atom of a substrate, resulting in a phosphoenzyme intermediate. In the case of PLDs a histidine nucleophile performs the first attack, whereas a catalytic threonine (T158) occurs in NPC4. Based on our structure and activity analyses (Fig. 2a, b), we propose that the T158 hydroxyl nucleophile is activated by neighboring high density of acidic residues (E23, D299 and E300) (Fig. 2c). A hydroxyl nucleophile has also been found in an acid phosphatase of *Francisella tularensis*, the AcpA[25]. In AcpA, a catalytic serine-bound metal ion was structural visualized in the active site[25], and a mechanism of the serine nucleophile activation by the metal ion was proposed. Supporting this, the EDTA presence could reduce AcpA activity by approximately 25%[35]. In the NPC4$^{1-496}$ structure, we also observed a metal ion in the active site (Supplementary Fig. 8a). Unlike AcpA, we found that the presence of EDTA had no effect on NPC4 activity (Supplementary Fig. 8c). Therefore, the metal ion in NPC4 is less likely to activate the hydroxyl nucleophile. However, even with EDTA concentration up to 5 mM in our activity assay system (using 1 μM enzyme), we cannot fully confirm that the structure-observed metal ion is depleted, and a possible activation mechanism of hydroxyl nucleophile by the metal ion cannot be excluded.

How NPC binds to phospholipid bilayer and dose this binding regulate NPC activity? Different from other phospholipase members, NPC encompasses only a conserved phosphoesterase domain (PD) and a previously uncharacterized C-terminal domain (CTD), lacking domains observed in other phospholipases required for calcium coordination, lipid binding, membrane association and activator regulation[12,13,26]. In this work, we characterize a C-terminal domain (CTD) of NPC4 that encompasses 3 α-helices (CTD$^{\alpha1}$, CTD$^{\alpha2}$ and CTD$^{\alpha3}$). Lots of work has showed that amphipathic helices are usually found in cytosolic enzymes and lipases, which bind to phospholipid bilayer and propagate this binding to enzyme activation[36–40]. Here in the NPC4$^{1-496}$ structure, we also find that the three helices of CTD are amphipathic helices (Supplementary Fig. 13). To evaluate if these amphipathic helices contribute to NPC4-bilayer association, we generated truncated NPC4 with different helix deletions and assessed their association to liposomes. Liposome sedimentation results showed that the deletion of CTD amphipathic helices had little effect on NPC4-liposome association (Supplementary Fig. 14a, b). Furthermore, we collected the pellet fraction of NPC4$^{1-496}$/liposome and measured its enzyme activity (Methods). We found that the activity of liposome-associated NPC4$^{1-496}$ was almost equal to that of free NPC4$^{1-496}$ (Supplementary Fig. 14c). Therefore, the amphipathic helices of CTD should contribute little to bilayer association and the bilayer binding does not regulate NPC4 activity. Instead, a role of CTD defined in our work is that this domain is indispensable for full activity of NPC4, via CTD$^{\alpha1}$-PD interaction to stabilize the active site in PD (Fig. 3 and Supplementary Fig. 9). Contribution of C-terminus to enzyme activity has also been characterized in the class of PLD[26,27]. However, unlike PLD, who harbors regulatory domains responsible for membrane association[28], NPC4 is tethered to lipid rafts in plasma membrane by a S-acylation occurred on a cysteine (Cys533) at the C terminus[23]. We thus surmise that the S-acylation proximal to CTD would dock PD onto the plasma membrane for substrates recognition/hydrolysis. Further investigation, including the structure determination and functional studies of S-acylated NPC4 in complex with membrane will be required to elucidate this aspect.

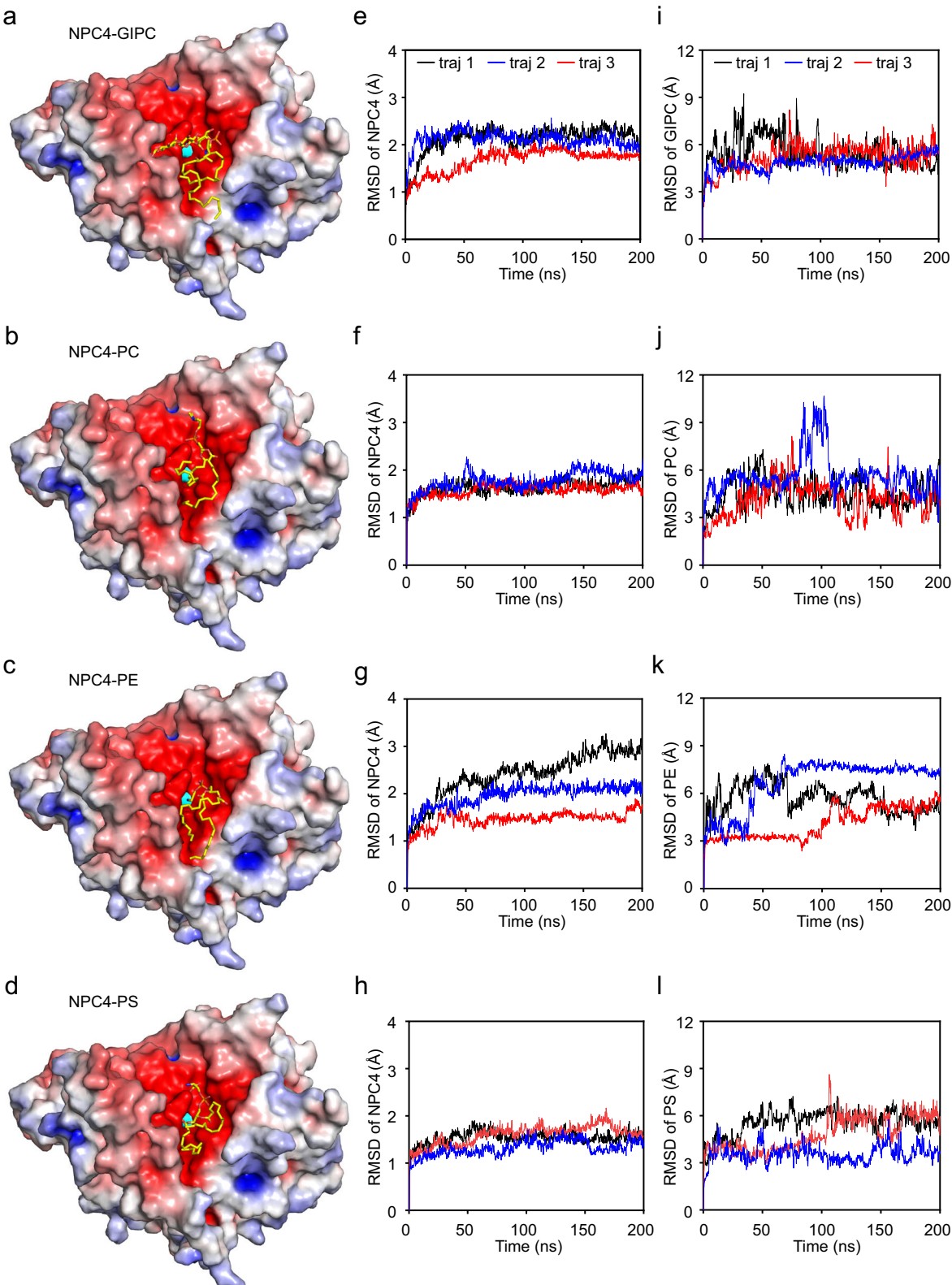

**Fig. 4 | Docked models of NPC4 in complex with various substrates and molecular dynamics simulations. a–d** Docked models of NPC4[1–496] in complex with GIPC, PC, PE and PS, respectively. Electrostatic surface of NPC4[1–496] is colored in terms of electrostatic potential, displayed in a scale from red (−5 kT/e) to blue (+5 kT/e). The catalytic residue T158 is colored in cyan, and substrates are shown in stick representation. **e–h** RMSD fluctuations of NPC4[1–496] in its docked models of GIPC, PC, PE and PS, respectively. Three independent MD trajectories (red, blue and black lines) of each docked model were performed, and the RMSD of NPC4[1–496] in the MD trajectory was calculated for all carbon atoms. And, **i–l** the RMSD fluctuation of the associated substrate in each model was also calculated, using all carbon atoms.

## Methods

### Protein expression and purification

The *NPC4* gene was amplified by PCR from the genomic DNA of Arabidopsis and subcloned into a pET15 vector with a N-terminal 6×His-tag. A DrICE protease cleavage site was constructed between the 6×His-tag and NPC4 for tag removal. A bonder of NPC4$^{1-496}$ (containing residues 1–496 and carrying K178A and K179A substitutions) was used for crystallization. Other NPC4 mutants used for biochemistry experiments were generated from NPC4$^{1-496}$. Protein was expressed in *E. coli* strain BL21(DE3) using Lysogeny broth (LB) medium. The cells were induced with 0.1 mM isopropyl-β-D-thiogalactoside (IPTG) at 16 °C for 12 h. Harvested cells were lysed by a high-pressure cell disrupter in a buffer containing 25 mM Tris-HCl pH8.0, 150 mM NaCl and 1 mM phenylmethanesulfonylfluoride (PMSF). Target protein was collected from the supernatant and purified over Ni$^{2+}$ affinity resin and HiTrap Q anion exchange column (GE Healthcare) used in tandem. The protein was further purified into homogeneity by gel-filtration chromatography (Superdex-200 Increase 10/300 GL, GE Healthcare) in a buffer containing 25 mM Tris-HCl pH8.0, 150 mM NaCl, and 5 mM dithiothreitol (DTT). Target fractions were collected for biochemistry experiments and crystallization.

To prepare selenomethionine (SeMet)-derived protein, transformed cells were cultured in SeMet-medium base containing 50 µg/mL L(+)−Selenomethionine. Protein expression and purification were performed in the same manner as the native protein.

### Crystallization and data collection

Crystallization experiments were carried out using the hanging-drop vapor-diffusion method at 18 °C by mixing equal volumes of protein and reservoir solution. Two sets of crystals were obtained. One of them were matured in about 7 days after crystallizing in a reservoir solution containing 0.1 M potassium phosphate monobasic (KH$_2$PO$_4$), 15% Glycerol and 18% PEG 8000 (condition A, enabled us to determine the structure of NPC4$^{1-496}$), and others were crystalized after about 2 months in a reservoir solution containing 0.2 M MgCl$_2$, 0.1 M HEPES, pH 7.2, 24 % PEG 3,350 and 0.1 M potassium sodium tartrate tetrahydrate (condition B, enabled us to determine the structure of NPC4$^{1-415}$). Our crystallization screening and optimization found that only condition A could enable the NPC4$^{1-496}$ crystal to mature with quality sufficient for structure determination. Cryo protection of crystals is 15% glycerol and 5% sucrose. Diffraction data were collected at a wavelength of 0.9897 Å on the Shanghai Synchrotron Radiation Facility (SSRF) beam lines BL19U1[41]. The phase of native protein was determined by Selenomethionine. SeMet-derived NPC4$^{1-496}$ was crystallized under the same condition as that of the native protein. The crystals were cryo-protected by serial transfers into reservoir solutions supplemented with 25% glycerol and then flash-cooled in liquid nitrogen. Data were collected at SSRF beamline BL17B, using the wavelength of 0.98 Å.

### Structure determination

The diffraction images were processed using XDS packages[42]. The phase of NPC4$^{1-496}$ was determined using the selenomethionine derived protein crystal. Further data processing was performed using CCP4 program suite[43]. Crystal structures of NPC4$^{1-415}$ and NPC4$^{1-496}$ were determined both at a resolution of 2.1 Å. All the structures were iteratively built with COOT[44] and refined using PHENIX program[45]. Data collection and structure refinement statistics were summarized in Supplementary Table 1. All figures were generated using the PyMOL program (http://www.pymol.org/). Electrostatic potential of protein is calculated with the APBS plugin in PyMOL.

### Activity assay

PC was used as the substrate for the activity assay of NPC4[8]. GIPC was extracted and purified from cabbage leaves according to a method described previously[46]. Briefly, 150–200 g of cabbage leaves were ground with 500 mL cold 0.1 N aqueous acetic acid. The slurry was filtered through 16 layers of acid-washed Miracloth and the filtrate was discarded. This step was repeated twice and the residue was then extracted with acidic 70% ethanol containing 0.1 N HCl (70 °C). The residue was filtered through Miracloth and washed with hot acidic 70%, then the filtrates were chilled immediately and put at −20 °C overnight. The sample was pelleted by centrifugation at 2000 *g* at 4 °C for 15 min. The pellet was successively washed with cold acetone and diethyl ether several times until yield a whitish precipitate. Then, the pellet was dissolved in THF/methanol/ water (4:4:1, v/v/v) containing 0.1% formic acid, followed by heating and sonication at 60 °C. After centrifugation at 2000 *g* at 4 °C for 10 min, the supernatant containing GIPC was retained and dried under a stream of N2. Then, butan-1-ol/water (1:1, v/v) was added for phase partition, and the upper butanolic phase contained GIPC. The GIPC extract was dried and the residue was dissolved in THF/methanol/water (4:4:1, v/v/v) containing 0.1% formic acid. The concentration of dissolved GIPC was quantified using LC-MS (6500 Plus QTRAP; SCIEX) with GM1 as standard. GIPC was suspended in 250 mM Tris-HCl pH 7.3 by sonication on ice for short time. Fifty microgram of protein (1 µM final concentration) was mixed with 100 µL of substrate solution in a final volume of 200 µL. The reaction mixture was incubated at 37 °C for 1 h, and stopped by adding of 200 µL of chloroform followed by vigorously vortexing. The lower layer was dried and dissolved in 50 µL THF/methanol/water (4:4:1, v/v/v) containing 0.1% formic acid for LC-MS/MS analyses. All the activity assay were performed with three biological replicates. All the used protein samples were stable in the assay system, confirmed by SDS-PAGE, as summarized in Supplementary Fig. 1. Our extracted GIPC substrate contained a small amount of GIPC-hydrolyzed product, hydroxyceramide. This contamination of GIPC substrate resulted in a detectable hydroxyceramide signal even in a blank assay (performing assay without enzyme). In the activity assay, we used hydroxyceramide signal to assess enzyme activity. For each measurement, we subtracted the blank hydroxyceramide signal and reported the relative activity of each mutant enzyme referenced to wild-type. The relative enzyme activity of some mutants with low GIPC-hydrolyzing ability were negative (−1% and −5% for H79A, −6% for T158E, and −8% for D299A), due to the blank hydroxyceramide signal subtraction. To avoid negative data points affect the statistics, we defined them with zero. Values of enzyme relative activity are means ± SD. *p* values were determined using unpaired two-tailed *t*-test. (NS, not significant, $^*p < 0.05$, $^{**}p < 0.01$, $^{***}p < 0.001$, $^{****}p < 0.0001$). The statistical analysis was performed using Graphpad Prism 8.0 software.

### Analysis of NPC4$^{1-496}$ phosphorylation by LC-MS/MS

PC4$^{1-496}$ crystallized in the presence of KH$_2$PO$_4$ was collected and subjected to in-solution digestion before MS analysis[47]. Sample was reduced with 5 mM dithiothreitol for 1 h at 37 °C and alkylated with 11 mM iodoacetamide for 30 min at room temperature in dark. The digested products were then diluted by adding 100 mM ammonium bicarbonate buffer to 1 ml. Finally, trypsin (Sequencing Grade Modified Trypsin, Promega) was added at 1:50 trypsin-to-protein mass ratio for the first digestion overnight and 1:100 trypsin-to-protein mass ratio for a second 4 h digestion at 37 °C. Tryptic peptides were then desalted with Strata X C18 SPE column (Phenomenex) and vacuum dried. Peptides were reconstituted in 20 µl 0.1% FA and separated by a 60 min gradient elution at a flow rate of 0.3 µl/min with the Thermo Easy-nLC1200 system, which was directly interfaced with a Q-Exactive HF hybridquadrupole Orbitrap. The raw data were analyzed by MaxQuant (version 1.6.14) using standard settings against the target protein sequence.

## Differential scanning fluorometry experiments

Real-time simultaneous monitoring of the intrinsic tryptophan fluorescence (ITF) at 330 nm and 350 nm of protein were carried out in a Prometheus NT.48 instrument (Nano Temper Technologies) with an excitation wavelength of 285 nm. 10 μM NPC4$^{1-496}$ was prepared in 25 mM Tris-HCl pH8.0 buffer containing 150 mM NaCl for DSF measurement. To verify the effect of phosphate on the protein stability, NPC4$^{1-496}$ was premixed with 50 mM KH$_2$PO$_4$ (pH8.0). The capillaries were filled with 10 μL sample and placed on the sample holder. A temperature gradient of 2 °C·min$^{-1}$ from 20 to 70 °C was applied and the intrinsic protein fluorescence emission at 330 and 350 nm was recorded.

## Limited proteolysis assay

About 20 μM NPC4$^{1-496}$ was incubated with trypsin or subtilisin protease at the serious protease concentrations (form 0.5 mg/mL to 0.23 μg/mL) at room temperature for 30 min. To assess the phosphate effect on the protein stability, NPC4$^{1-496}$ was premixed with 50 mM KH$_2$PO$_4$ (pH8.0) at 4 °C for 30 min and then transferred for protease digestion. Digestion reactions were stopped by adding 100 mM PMSF. Digested products were analyzed by SDS-PAGE and visualized by coomassie blue staining.

## Model docking and molecular dynamics simulations of NPC4$^{1-496}$–substrate complex

We used the program AutoDock Vina 1.1.2[34] to dock the complex models of NPC4$^{1-496}$ and various substrate molecules, including GIPC, PC, PS and PE. The original structure of these lipid molecules were generated or modified using Pymol and further prepared with Auto-Dock tools[48] for the docking process. For NPC4$^{1-496}$ used in docking, the observed phosphate, metal ion and T158-linked phosphate in NPC4$^{1-496}$ crystal structure were removed, and the missing residues (56–59) were built using Pymol and further refined with Xplor-NIH[49]. The NPC4$^{1-496}$ structure was treated as a rigid body, and the ligands were given full torsion freedom during the docking calculation. The docking box was set around the catalytic area. The center of docking box was set at the oxygen side-chain atom of T158 with box size of 40 Å×40 Å×40 Å. A total of 10 models were generated for each substrate docking and the model with best binding affinity was selected for the further analysis. The binding affinity for these substrates were: −4.9 kcal/mol for GIPC, −5.2 kcal/mol for PC, −5.1 kcal/mol for PS, and −5.0 kcal/mol for PE, respectively. Furthermore, we also performed MM-GBSA[50] to calculate the binding free energy for these docked complexes. The binding free energy were −41.41 kcal/mol for GIPC, −33.63 kcal/mol for PC, −62.68 kcal/mol for PS, and −46.24 kcal/mol for PE, respectively. These analyses thus indicate that the docked models of enzyme-substrate should be stable.

The molecular dynamics (MD) simulations for all the NPC4$^{1-496}$-substrate models were performed with AMBER 16 software[51]. The complex structures with best binding affinity were used as the starting conformation. The Amber ff14SB[52] and GAFF[53] force field were used for the protein and substrate, respectively. The whole system was solvated in a cubic TIP3P water box with a 10 Å padding for every direction. We have performed three independent trajectories with different random seeds for each NPC4$^{1-496}$-substrate complex. The production process lasts 200 ns at 298 K with a time step of 2 fs. The CPPTRAJ module in the AMBER 16 package was used to calculate the root-mean-square deviation (RMSD) for all carbon atoms.

## Liposome sedimentation assay

Liposome were generated using Soy PC (Avanti Polar Lipids 840054 P). 1 mM liposome and 10 μM protein were mixed in 25 mM Tris-HCl pH8.0 buffer containing 150 mM NaCl, and incubated on a rotation mixer at 18 °C for 1 h. The liposome/protein mixture was centrifuged at 110,000 g at 4 °C for 1 h. The supernatant fraction was collected and

the pellet fraction was resuspended in an equal volume for SDS-PAGE analysis. The SDS-PAGE gel bands were quantified using ImageJ[54]

## Reporting summary

Further information on research design is available in the Nature Portfolio Reporting Summary linked to this article.

## Data availability

Atomic coordinates of NPC4$^{1-415}$ and NPC4$^{1-496}$ have been deposited in the Protein Data Bank (PDB) under accession number 8HAV and 8HAW, respectively. Primers used in this study are listed in Supplementary Dataset 1. Source data are provided with this paper.

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

## Acknowledgements

We thank the staffs from BL17B/BL19U1 beamline of National Facility for Protein Science in Shanghai (NFPS) at Shanghai Synchrotron Radiation Facility for assistance during data collection, and Dr. Delin Zhang at the Center for Protein Research at Huazhong Agricultural University. This work has been supported by the National Key R&D Program of China (2018YFA0507700 to Z.L., Z.G. and P.Y.), the National Natural Science Foundation of China (32071226 to Z.L.), the Foundation of Hubei Hongshan Laboratory (2021HSZD004 to Z.L., and 2021HSZD011 to L.G.), the HZAU-AGIS Cooperation Fund (SZYJY2022022 to Z.L.), and the USDA National Institute of Food and Agriculture [2020-67013-30908/project accession number 1022148 to X.M.W.]. Z.G. was supported by Youth Innovation Promotion Association of the Chinese Academy of Sciences (2020329).

## Author contributions

L.G., and Z.L. conceived the project. R.Y.F., F.Z., Z.G., Y.K.C., B.Y., C.Z., J.Z., Z.M.D., X.M.W., and P.Y. designed all experiments and analyzed data. R.Y.F. performer crystallization and biochemical experiments. F.Z. resolved the structures. Z.G. performed the docking and MD simulations. R.Y.F., L.G., and Z.L. wrote the manuscript with support from all authors.

## Competing interests

The authors declare no competing interests.
