## [Peer Review File · Nature Communications]

Insights into the mechanism of phospholipid hydrolysis by plant non-specific phospholipase CEditorial Note: This manuscript has been previously reviewed at another journal that is not operating a transparent peer review scheme. This document only contains reviewer comments and rebuttal letters for versions considered at Nature Communications.

REVIEWER COMMENTS

Reviewer #1 (Remarks to the Author):

The revised manuscript has been significantly improved, and most concerns have been addressed.

Two minors:

- 1. If phosphorylation and metal ion are not presented in the truncated catalytic domain structure, it should be described clearly in the Report section.**
- 2. The discussion is very concise. More discussion of the hypothesis with a cartoon model based on the structures will better inform readers of the proposed activation mechanism.**

Reviewer #2 (Remarks to the Author):

Fan et al present here the crystal structures of Non-specific phospholipase C. Their crystallographic data combined with functional assays provide evidence that the C-terminal domain (CTD) and its interaction with phosphoesterase domain (PD) is crucial for catalysis. They identified the mechanism of hydrolysis to be similar to other phospholipases and involve the formation a phosphorylated transition state. This paper is a resubmission of an earlier manuscript where the authors had two crystal structures and smFRET data to observe the activation mechanism. Based on the extensive comments they removed the smFRET data and claims therein and re-prepared the crystals for conditions different from the ones used for the active conformation.

Identifying the structure and the mechanism of activation of a phospholipase can be interesting for the community. The mechanism of hydrolysis the authors now present, however is of limited novelty. As such this work might be more suitable for a more specialized journal.

A few elements to consider prior to resubmission .

- 1) Authors crystallized under "another crystallizing condition" NPC and found a structure with no electron density for CTD . Then they expresses a protein that lacks the CTD and found its inactive and thus conclude that CTD is required for catalysis. The direct correlation is not fully supported by the provided data. How do authors exclude that degradation (as they write in the response to their previous submission comments) which could be partial has the same effect as completely omitting the CTD ?**
- 2) Author propose that NPC4 follows a mechanism similar to the "classical" lipase function mechanism where tetrahedral intermediates are stabilized by hydrogen bonds or/and charges. Authors claim that in their structure the position of phosphate group on the modified T158 overlaps with the orthovanadate inhibitor observed in AcpA, however they provide no structure with inhibitor bound to the active site, an instrumental element for deciphering the roles of each of the catalytic acid amino acids.**
- 3) Why is KH₂PO₄ used for the first structure, especially as it seems to bind at the entrance of the catalytic pocket ? similarly why is there degradation of the NPC in the 2nd buffer conditions ? authos do not discuss this or not try to find conditions that degradation is not present.**
- 4) Authors provide autodock Vina structures of NPC41-496 and substrates, as they could not attain crystal structures. The exposed cleft is larger than the molecular size of the tested substrates GIPC or PC, and the head group of these two substrates insert toward the catalytic pocket from two directions, an element the authors attribute to promiscuity. How**

stable are these docking sites ? what are the docking scores ? do additional substrates with diverse structures also bind in a similar way ? are these supported by simulations ?

5) Did authors evaluate whether the three alpha helices of NPC are amphipathic helices ?. there is extensive work on protein in general and lipases phospholipases specifically showing such AH helices to bind to bilayers, an element recently show to activate lipases. Did authors examine whether any of the helices is amphipathic? (Bruno Antony has a online tool for this named helliquest)

6) figures 2b, 3a do not have P test values

Reviewer #3 (Remarks to the Author):

The manuscript by Fan and Zhao et al. attempts to characterize a non-specific phospholipase C (NPC) structurally. NPC represents a novel type of plant phospholipid-cleaving enzymes producing a crucial signalling molecule, diacylglycerol. Members of the NPC family were implicated in various essential biological processes, including root growth, gametophyte development or plant response to multiple stresses. Contrary to other phospholipases (PLA1, PLA2, PLD and PI-PLC), structural aspects of the NPC function are unknown. The authors of the submitted manuscript report the structure of Arabidopsis NPC4, revealed by X-ray crystallography and resolved to 2.1 Å resolution, thus providing the first structural insight into this essential enzyme. The authors show that NPC is structurally a unique phospholipase distinct from other experimentally characterized phospholipases. Instead, the NPC structure is homologous to the acid phosphatase A (AcpA) despite a low sequence identity and different substrate preferences of these two enzymes. NPC4 is divided into two domains, a phosphoesterase domain (PD) and a previously uncharacterized C-terminal domain (CTD). The authors convincingly show that CTD contributes to NPC activity by stabilizing the catalytical pocket located in PD. Using a series of site-directed mutations, the authors suggest a unique catalytical mechanism for the NPC enzymes. However, the proposed mechanism of the NPC function is highly speculative and is not supported by the data. Moreover, the results are only superficially discussed in the manuscript.

Major points:

1)The authors describe that the critical residues in the catalytical pocket of NPC4 adopt identical conformation to the catalytical residues of AcpA. The authors also show that, similarly to AcpA, NPC4 contains a metal ion in its catalytical pocket. However, the authors did not observe any effect of EDTA on the enzyme activity. Given that a mutation of any NPC4 residue coordinating the metal ion strongly impacts the activity, the authors should demonstrate that the metal ion is indeed depleted after adding EDTA. Do the mutants in the metal ion-coordinating residues lead to ion depletion? How would EDTA affect NPC4 activity if it is used already during the protein purification?

2)The authors suggest that NPC4 uses a different catalytical mechanism than AcpA. Given the almost identical active sites, I would argue that the opposite is the case, i.e. the enzymes share the mechanism of their function. The scheme of the proposed chemical mechanism (Supplementary Figure 5) contains many unorthodox steps and is highly

speculative. The scheme has arbitrarily appearing and disappearing protons or amino acid residues. R' should represent a diacylglycerol moiety and not fatty acyl chains.

3) The authors convincingly show that T158 is phosphorylated (Supplementary Figure 5). In the case of phospholipase D, the presence of phosphohistidine without a substrate was suggested to reflect an autoinhibited state of the enzyme (Bowling et al., 2020, 10.1038/s41589-020-0499-8). How is NPC activity affected by a phospho-mimetic mutation of T158? Such a mutation would add to the elucidation of the NPC4 catalytical mechanism.

4) The authors state that their substrate-protein docking results indicate the mechanism for the NPC4 promiscuous activity. The authors should discuss how the substrates' different locations affect the cleaved bond's position relative to the catalytical residues. How do the docking results fit with the proposed catalytical mechanism? How would be the substrate binding affected by the presence of the phospholipid bilayer?

Minor points:

1) The last paragraph of the discussion part (GIPC, ...) is vague, and as currently written, it is irrelevant to the story.

2) In Figure 2 and Supplementary Figure 3, close-ups of the active sites are deliberately rotated, which is somewhat confusing.

3) The authors provide the coordinates and box sizes for the ligand-protein docking calculations. However, as currently described, it is difficult to assess which part of the protein was actually used. The authors might want to provide the figures depicting the selected areas on the protein.

REVIEWER COMMENTS

Reviewer #1 (Remarks to the Author):

The revised manuscript has been significantly improved, and most concerns have been addressed.

Response: Thank you for your thoughtful and constructive comments in revising our previous manuscript. We are delighted that you found our manuscript largely improved.

Two minors:

1. If phosphorylation and metal ion are not presented in the truncated catalytic domain structure, it should be described clearly in the Report section.

Response: Thank you for pointing this out. We have clearly described these differences in revision (paragraph 1 in page 8).

2. The discussion is very concise. More discussion of the hypothesis with a cartoon model based on the structures will better inform readers of the proposed activation mechanism.

Response: Thank you for these suggestions. In the revised manuscript, we have added a Discussion paragraph (paragraph 3 in page 10) to compare the catalytic mechanism of NPC4 with other phospholipase members, and to highlight the new mechanistic insights into the phospholipases provided by this study. Regarding to your suggestion of a cartoon model for proposing NPC4 activation mechanism, as we focus this study on the structural and mechanistic understanding of phospholipid-hydrolyzing by NPC4 and we have no solid data to demonstrate an activation mechanism, we wouldn't like to include a cartoon model. Related to the regulation of NPC4, in revision we have added a paragraph to analyze the possible bilayer/NPC4 interaction and the regulation of NPC4 by the bilayer binding (paragraph 2 in page 11).

Reviewer #2 (Remarks to the Author):

Fan et al present here the crystal structures of Non-specific phospholipase C. Their crystallographic data combined with functional assays provide evidence that the C-terminal domain (CTD) and its interaction with phosphoesterase domain (PD) is crucial for catalysis.

They identified the mechanism of hydrolysis to be similar to other phospholipases and involve the formation a phosphorylated transition state. This paper is a resubmission of an earlier manuscript where the authors had two crystal structures and smFRET data to observe the activation mechanism. Based on the extensive comments they removed the smFRET data and claims therein and re—prepared the crystals for conditions different from the ones used for the active conformation.

Identifying the structure and the mechanism of activation of a phospholipase can be interesting for the community. The mechanism of hydrolysis the authors now present, however is of limited novelty. As such this work might be more suitable for a more specialized journal.

Response: While we fully agree with you that unravelling the activation mechanism of a phospholipase is interesting, solving the structural basis and catalytic mechanism of NPCs are also important aspects to help us better understand the function of these fundamental plant enzymes. Since the discovery of NPCs, they have gained a lot of interest. This is because they are a novel type of plant phospholipases and play various fundamental roles in plant growth and development. Moreover, recently papers in the area have found that NPC4 can help plant to cope with phosphate limitation (DOI: 10.1093/plcell/koaa054, DOI: 10.1111/tbj.15260). Therefore, while the interest of NPCs are clear, we do not know how they really work. And, among various classes of phospholipases (including PLA1, PLA2, PLD, PI-PLC, and NPC), eukaryotic NPC is the only one whose structure and working mechanism have remained uncharacterized. In this manuscript, we solve this mystery and define the molecular basis of how NPC4 works, and provide new mechanistic insights into the members of phospholipase family. We humbly think that this work deserves publication in this journal.

We truly appreciate your insightful comments and very constructive suggestions that helped us to significantly improve our revised manuscript. According to your suggestions, we have performed additional experiments and analyses. These results and discussions were included in the revised manuscript. Please find our point-by-point response to your concerns listed below.

A few elements to consider prior to resubmission .

- 1) Authors crystallized under “another crystallizing condition” NPC and found a structure with no electron density for CTD. Then they expresses a protein that lacks the CTD and found its

inactive and thus conclude that CTD is required for catalysis. The direct correlation is not fully supported by the provided data. How do authors exclude that degradation (as they write in the response to their previous submission comments) which could be partial has the same effect as completely omitting the CTD ?

Response: Thank you for pointing this out. In revision, we examined all the used protein samples to check whether they were degraded after performing the substrate-hydrolysis reaction in activity assays. SDS-PAGE results showed that all the samples are stable in the assay system. Thus, our observed activity changes of NPC4 mutants are rarely caused by protein degradation. We have included these SDS-PAGE results in revision and added a figure as Supplementary Fig. 1 in the revised Supplementary Information. In addition, we have included the degradation result of NPC4¹⁻⁴¹⁵ structure crystallized under another condition and related discussion (paragraph 1 in page 8, and Supplementary Figs. 10 and 11), and re-organized the “**CTD contributes NPC4 activity via CTD^{a1}-PD interaction**” result, to make our conclusion more convincing.

2) Author propose that NPC4 follows a mechanism similar to the “classical” lipase function mechanism where tetrahedral intermediates are stabilized by hydrogen bonds or/and charges. Authors claim that in their structure the position of phosphate group on the modified T158 overlaps with the orthovanadate inhibitor observed in AcpA, however they provide no structure with inhibitor bound to the active site, an instrumental element for deciphering the roles of each of the catalytic acid amino acids.

Response: While we completely agree with you that a structure of NPC4 in complex with a bound inhibitor would help to confirm the catalytic residues, we can propose the roles of the catalytic residues based on our NPC4¹⁻⁴⁹⁶ structure and site-directed mutation evidences, as reasoned below:

Firstly, the PD domain of NPC4¹⁻⁴⁹⁶ folds similar to AcpA, as well as the key residues for catalysis in their catalytic pockets. It was previously showed in AcpA that the S175 hydroxyl nucleophile performs the first attack onto the phosphorus atom of a substrate. Here we find that the residue T158 of NPC4 is structurally conserved with S175 in AcpA. We thus infer that the oxygen side-chain atom of T158 performs the first nucleophilic attack in substrate-hydrolyzing by NPC4. Supporting this, alanine substitution of T158 abolishes NPC4 activity.

Secondly, the phosphate group modified on T158 of NPC4 overlaps with the orthovanadate inhibitor bound to S175 in AcpA, suggesting this covalent-linked phosphate group would inhibit NPC4 activity. In addition, a phosphorylation on the catalytic residue has also been previously observed in PLDs without a substrate, wherein the phosphorylation was suggested to block the nucleophilic atom to initiate the first attack. Consistent with this, in revision we found that a phospho-mimetic mutation of T158 (T158E or T158D) in NPC4 eliminates the enzyme activity. We thus propose that the residue T158 acts as the first nucleophile. We have included these new results of phospho-mimetic mutation and discussion in the revised manuscript (paragraph 1 in page 6).

Thirdly, in revision we further identified another residue of NPC4, the D76, that is also important for enzyme activity. And, together with our other site-directed mutation results and structure analyses, we have carefully revised our scheme of the chemical mechanism for the substrate-hydrolyzing by NPC4. We have added these mentioned things in the revised manuscript (paragraph 2 in page 7).

3) Why is KH_2PO_4 used for the first structure, especially as it seems to bind at the entrance of the catalytic pocket ? similarly why is there degradation of the NPC in the 2nd buffer conditions ? authors do not discuss this or not try to find conditions that degradation is not present.

Response: We are sorry for missing these information.

KH_2PO_4 was inherently presented in the commercial crystallizing reservoir solution (RIGAKU), not by manually adding. For the crystallization, two sets of crystals were obtained. One of them were matured in about 7 days after crystallizing in a reservoir solution containing 0.1 M potassium phosphate monobasic (KH_2PO_4), 15% Glycerol and 18% PEG 8000 (condition A, enabled us to determine the structure of NPC4¹⁻⁴⁹⁶), and others were crystallized after about 2 months in a reservoir solution containing 0.2 M MgCl_2 , 0.1 M HEPES, pH 7.2, 24 % PEG 3,350 and 0.1 M potassium sodium tartrate tetrahydrate (condition B, enabled us to determine the structure of NPC4¹⁻⁴¹⁵). Our crystallization screening and optimization found that only condition A could enable the NPC4¹⁻⁴⁹⁶ crystal to mature with quality sufficient for structure determination. We have included these information in the revised Methods.

We found that the CTD omitting in the structure of NPC4¹⁻⁴¹⁵ should be due to protein

degradation during the time-consuming crystallization under condition B, while the crystal of NPC4¹⁻⁴⁹⁶ remains almost intact in condition A (Supplementary Fig. 10 in revised Supplementary Information). Different scanning fluorimetry (DSF) analysis and limited proteolysis experiments revealed that the presence of KH₂PO₄ can enhance the stability of NPC4. We have included these results and explanation in the revised manuscript (paragraph 1 in page 8), and added a figure as Supplementary Fig. 11.

4) Authors provide autodock Vina structures of NPC4¹⁻⁴⁹⁶ and substrates, as they could not attain crystal structures. The exposed cleft is larger than the molecular size of the tested substrates GIPC or PC, and the head group of these two substrates insert toward the catalytic pocket from two directions, an element the authors attribute to promiscuity. How stable are these docking sites? what are the docking scores? do additional substrates with diverse structures also bind in a similar way? are these supported by simulations?

Response: Thank you for these comments and constructive suggestions. In revision, we have performed the molecular docking with additional substrates, supplemented with molecular dynamics simulations and provided additional analysis and discussion, as shown below:

We have carefully re-performed docking and presented the docking details in the revised Methods. In revision, we docked the complex models of NPC4¹⁻⁴⁹⁶ and various substrates, including GIPC, PC, PE and PS. A total of 10 models were generated for each substrate docking and the model with best binding affinity was selected for the further analysis. The binding affinity for these substrates were: -4.9 kcal/mol for GIPC, -5.2 kcal/mol for PC, -5.0 kcal/mol for PE and -5.1 kcal/mol for PS, respectively. Furthermore, we also performed MM-GBSA to calculate the binding free energy for these docked complexes. The binding free energy were -41.41 kcal/mol for GIPC, -33.63 kcal/mol for PC, -46.24 kcal/mol for PE and -62.68 kcal/mol for PS, respectively. Thus, these analyses indicate that the docked models of enzyme-substrate should be stable. We have included these information in the revised Methods.

These docked complexes show that the four substrates are targeted into the funneled-negative charged cleft leading to the catalytic pocket, and the head groups point toward the catalytic pocket in alternative positions. Furthermore, in revision we performed multiple rounds of MD simulations on these docked models. Three independent MD trajectories were run for each

docked model and results showed that these models are stable. Based on these docked enzyme-substrate models, we analyzed whether the conformations are competent for catalysis. As the distances between the nucleophilic atom of NPC4 and the phosphorus atom of substrates are larger than that required for an efficient nucleophilic attack, we thus propose that the docked conformation of a substrate should represent an initial-recognized state, and the substrate is about to be pulled into the catalytic pocket for cleavage, orchestrated by some enzyme conformational changes. We have included these results and discussion in our revised manuscript (paragraph 2 in page 9 and paragraph 1 in page 10) and Supplementary Fig. 12.

5) Did authors evaluate whether the three alpha helices of NPC are amphipathic helices?. there is extensive work on protein in general and lipases phospholipases specifically showing such AH helices to bind to bilayers, an element recently show to activate lipases. Did authors examine whether any of the helices is amphipathic? (Bruno Antony has a online tool for this named helliquest)

Response: Thank you for these constructive suggestions. In revision, we evaluated that the three α -helices of CTD are amphipathic helices. To asses whether these AH helices contribute to NPC4-bilayer association and whether this binding regulates NPC activity, we performed additional experiments.

We generated truncated NPC4 with different helix deletions and assessed their association to liposomes. Liposome sedimentation results showed that the deletion of CTD amphipathic helices had little effect on NPC4-liposome association. Furthermore, we collected the pellet fraction of NPC4¹⁻⁴⁹⁶/liposome and measured its enzyme activity. We found that the activity of liposome-associated NPC4¹⁻⁴⁹⁶ is almost equal to that of free NPC4¹⁻⁴⁹⁶. Therefore, the amphipathic helices of CTD should contribute little to bilayer association and the bilayer binding does not regulate NPC4 activity. We have included this results and discussion in the revised manuscript (paragraph 2 in page 11) and Supplementary Figs. 13, 14.

6) figures 2b, 3a do not have P test values

Response: Thank you. We have added these values.

Reviewer #3 (Remarks to the Author):

The manuscript by Fan and Zhao et al. attempts to characterize a non-specific phospholipase C (NPC) structurally. NPC represents a novel type of plant phospholipid-cleaving enzymes producing a crucial signalling molecule, diacylglycerol. Members of the NPC family were implicated in various essential biological processes, including root growth, gametophyte development or plant response to multiple stresses. Contrary to other phospholipases (PLA1, PLA2, PLD and PI-PLC), structural aspects of the NPC function are unknown. The authors of the submitted manuscript report the structure of Arabidopsis NPC4, revealed by X-ray crystallography and resolved to 2.1 Å resolution, thus providing the first structural insight into this essential enzyme.

The authors show that NPC is structurally a unique phospholipase distinct from other experimentally characterized phospholipases. Instead, the NPC structure is homologous to the acid phosphatase A (AcpA) despite a low sequence identity and different substrate preferences of these two enzymes. NPC4 is divided into two domains, a phosphoesterase domain (PD) and a previously uncharacterized C-terminal domain (CTD). The authors convincingly show that CTD contributes to NPC activity by stabilizing the catalytical pocket located in PD. Using a series of site-directed mutations, the authors suggest a unique catalytical mechanism for the NPC enzymes. However, the proposed mechanism of the NPC function is highly speculative and is not supported by the data. Moreover, the results are only superficially discussed in the manuscript.

Response: We are delighted that you recognize the importance of our work. We truly appreciate your insightful comments and very constructive suggestions that helped us to significantly improve our manuscript. According to your suggestions, we have performed additional experiments and analyses. These results and discussions were included in the revised manuscript. Please find our point-by-point response to your concerns listed below.

Major points:

1)The authors describe that the critical residues in the catalytical pocket of NPC4 adopt identical conformation to the catalytical residues of AcpA. The authors also show that, similarly to AcpA, NPC4 contains a metal ion in its catalytical pocket. However, the authors did not observe any effect of EDTA on the enzyme activity. Given that a mutation of any NPC4 residue coordinating the metal ion strongly impacts the activity, the authors should demonstrate that the metal ion is indeed depleted after adding EDTA. Do the mutants in the metal ion-coordinating residues lead to ion depletion? How would EDTA affect NPC4 activity if it is used already during the protein purification?

Response: Thank you for these comments and constructive suggestions.

According to your suggestions, we further performed additional experiments and analyses. In revision, to clarify if this co-crystallized metal ion is involved in substrate hydrolysis, we prepared NPC4 in the presence of ethylenediaminetetraacetic acid (EDTA) or divalent metal ions, and assessed their effect on the enzyme activity. By adding 2 mM EDTA into the activity assay buffer or maintaining 2 mM EDTA during the protein purification, for chelating potential divalent metal ions in protein, we observed no changes in NPC4 activity compared to that in the absence of EDTA. Furthermore, increasing the added EDTA to a higher concentration of 5 mM still did not reduce the enzyme activity, indicating no involvement of metal ions in catalysis. In parallel, we speculated that if a metal ion is involved in catalysis, adding ions to the assay buffer would increase NPC4 activity. By adding 1 mM divalent metal ions, such as Ca^{2+} , Mg^{2+} or Zn^{2+} , into the activity assay buffer, we found that the enzyme activity did not increase, suggesting that these ions are not required for NPC4 catalysis. The presence of Zn^{2+} partially reduced the enzyme activity, possibly due to protein instability caused by Zn^{2+} or other unknown reasons. Inhibition of Zn^{2+} on NPC4 activity was also found in previous reports. Together, it is most likely that the observed metal ion in our crystal structure is unnecessary for NPC4 activity, consistent with previous findings that the NPC4 activity is ion independent. We have included these results and discussion in the revised manuscript (paragraph 3 in page 6 and paragraph 1 in page 7) and Supplementary Fig. 8c.

As we have not directly confirm that the structure-observed metal ion is depleted by the presence of EDTA, a possible activation mechanism of hydroxyl nucleophile by the metal ion cannot be excluded. In the revised manuscript, we have added a Discussion paragraph (paragraph 3 in page

10 and paragraph 1 in page 11) to discuss this possibility, and to compare the catalytic mechanism of NPC4 with AcpA and other phospholipase members.

2)The authors suggest that NPC4 uses a different catalytical mechanism than AcpA. Given the almost identical active sites, I would argue that the opposite is the case, i.e. the enzymes share the mechanism of their function. The scheme of the proposed chemical mechanism (Supplementary Figure 5) contains many unorthodox steps and is highly speculative. The scheme has arbitrarily appearing and disappearing protons or amino acid residues. R' should represent a diacylglycerol moiety and not fatty acyl chains.

Response: Thank you for pointing these out.

In this work, we found that although the active site of NPC4 is structurally similar to that of AcpA, it is stabilized by the extra CTD domain in NPC4, which doesn't occur in AcpA. To make our statement clearer and more accurate, in revision we replaced "Regarding the similar enzyme structures targeting at different substrates for hydrolysis, we thus infer NPC4 would use different mechanism for functioning." (lines 121-123 in our previous manuscript) by using "As these similar phosphoesterase domain structures take different substrate preferences, we infer that NPC4 would use a different mechanism for targets recognition and / or hydrolysis, wherein the extra CTD domain in NPC4 is indispensable (see below)." (paragraph 1 in page 5)

Regarding to your comments and suggestion on the catalytic mechanism of NPC4, in revision we further performed additional experiments and analyses (including, paragraphs 1 and 3 in page 6, and paragraph 1 in page 7). And, based on our experimental results and structure analyses, we have carefully revised our scheme of the chemical mechanism for the substrate-hydrolyzing by NPC4 (paragraph 2 in page 7). Moreover, we have added a Discussion paragraph (paragraph 3 in page 10 and (paragraph 1 in page 11) to compare the catalytic mechanism of NPC4 with AcpA and other phospholipase members.

3)The authors convincingly show that T158 is phosphorylated (Supplementary Figure 5). In the case of phospholipase D, the presence of phosphohistidine without a substrate was suggested to reflect an autoinhibited state of the enzyme (Bowling et al., 2020, 10.1038/s41589-020-0499-8). How is NPC activity affected by a phospho-mimetic mutation of T158? Such a mutation would add to the elucidation of the NPC4 catalytical mechanism.

Response: Thank you for pointing this out. In revision, we included a discussion about the autoinhibited state of the phosphoenzyme (paragraph 1 in page 6). Moreover, we constructed a phospho-mimetic mutation of T158 (T158E or T158D) and results showed that this mutation eliminates NPC4 activity. Therefore, in the case of NPC4 it should be the oxygen side-chain atom of T158 that act as nucleophilic atom to initiate the first attack, and consequently its phosphorylation would block the substrate-hydrolysis activity of NPC4. We have included these new results of phospho-mimetic mutation and discussion in the revised manuscript (paragraph 1 in page 6). Thank you for these constructive suggestions that helped up to elucidate the NPC4 catalytic mechanism (paragraph 2 in page 7).

4) The authors state that their substrate-protein docking results indicate the mechanism for the NPC4 promiscuous activity. The authors should discuss how the substrates' different locations affect the cleaved bond's position relative to the catalytical residues. How do the docking results fit with the proposed catalytical mechanism? How would be the substrate binding affected by the presence of the phospholipid bilayer?

Response: Thank you for your comments and constructive suggestions.

To supporting the indicated mechanism for the NPC4 promiscuous activity, in revision we further performed the molecular docking of NPC4¹⁻⁴⁹⁶ with more substrates, including GIPC, PC, PE and PS, and ran extensive molecular dynamics simulations of these docked NPC4-substrate models. Docking results showed that the four substrates are targeted into the funneled-negative charged cleft leading to the catalytic pocket, and the head groups point toward the catalytic pocket in alternative positions. And, these docked enzyme-substrate models are stable, supported by MD simulations. We have included these new results and discussion in the revised manuscript (paragraph 2 in page 9).

Based on these docked enzyme-substrate models, we analyzed whether the conformations are competent for catalysis. As the distances between the nucleophilic atom of NPC4 and the phosphorus atom of substrates are larger than that required for an efficient nucleophilic attack, we thus propose that the docked conformation of a substrate should represent an initial-recognized state, and the substrate is about to be pulled into the catalytic pocket for cleavage, orchestrated by some enzyme conformational changes. We have included these results

and discussion in our revised manuscript (paragraph 1 in page 10) and Supplementary Fig. 12.

In revision, although we couldn't quantify the changes of substrate binding affinity induced by the presence of the phospholipid bilayer, we assessed whether the NPC4-bilayer binding regulates enzyme activity. We incubated NPC4¹⁻⁴⁹⁶ with constituted liposomes, collected the pellet fraction of NPC4¹⁻⁴⁹⁶/liposome by liposome sedimentation and measured its enzyme activity. We found that the activity of liposome-associated NPC4¹⁻⁴⁹⁶ is almost equal to that of free NPC4¹⁻⁴⁹⁶. Therefore, the bilayer binding does not regulate NPC4 activity. We have included these results and discussion in the revised manuscript (paragraph 2 in page 11) and Supplementary Figs. 13, 14.

Minor points:

1) The last paragraph of the discussion part (GIPC, ...) is vague, and as currently written, it is irrelevant to the story.

Response: Thank you for pointing this out. We have deleted this in revision.

2) In Figure 2 and Supplementary Figure 3, close-ups of the active sites are deliberately rotated, which is somewhat confusing.

Response: We are sorry for this confusion. We have revised the representations and kept them in a same projection.

3) The authors provide the coordinates and box sizes for the ligand-protein docking calculations. However, as currently described, it is difficult to assess which part of the protein was actually used. The authors might want to provide the figures depicting the selected areas on the protein.

Response: We are sorry for missing details.

For docking, The NPC4¹⁻⁴⁹⁶ structure was treated as a rigid body, and the ligands were given full torsion freedom during the docking calculation. The docking box was set around the catalytic area. The center of docking box was set at the oxygen side-chain atom of T158 with box size of 40 Å×40 Å×40 Å (Response Fig. 1). We have carefully revised and clearly described the docking methods in the revised Methods (please see Methods for the full details).

Response Fig. 1. A box size of $40 \text{ \AA} \times 40 \text{ \AA} \times 40 \text{ \AA}$ is centered at the oxygen side-chain atom of T158 in NPC4¹⁻⁴⁹⁶. The oxygen atom is shown in red sphere representation.

REVIEWERS' COMMENTS

Reviewer #2 (Remarks to the Author):

The authors went into great depths to address all my technical comments. This rectified all the unclear experimental elements that were raising concerns on the experimental rigor. The authors presented now even more solid data supporting that NPC operates in similar way to other phospholipase. My concern on the novelty of the work is therefore partially solved but can recommend publication of this novelty they describe in the rebuttal is inserted in the main text

Reviewer #3 (Remarks to the Author):

The authors have largely addressed all my concerns in the revised version of the manuscript. Nevertheless, I have several comments on the novel results provided by the authors.

- 1)The authors performed all-atom molecular dynamics simulations of the NPC4 enzyme with different substrates. In Figure 4, the authors report RMSD values over the performed simulations to indicate stable models. RMSD reflects on the stability of either the enzyme or the substrate but less on the stability of the substrate-enzyme complex. The authors should show the distance between the substrate and, e.g. T158 as a function of time to assess the complex's stability.
- 2)In Figure 2b showing the NPC4 relative activity, some data points are either missing or are negative. How could be the enzyme activity negative? How does this fact affect the statistics shown?
- 3)The authors state that Zn²⁺ ions partially reduce the NPC4 activity. However, Supplementary Fig. 8c shows a partial reduction of the activity in the presence of Mg²⁺ and not Zn²⁺.
- 4)It is unclear if the experiment shown in panel a of Supplementary Fig. 14 results from one or several replicas. The authors should attempt to quantify the liposome binding and the effect of truncations on binding. How would liposome association be affected by the presence of negatively charged phospholipids which are hallmarks of the plasma membrane?
- 5)Newly added text contains several typos, and some sentences are hard to follow, e.g. "autoinhibited sate" (page 6), "a catalytic mechanism of substrate-hydrolysing" (page 7), "densities is" (page 8), "aside from the catalytic pocket which concerning with" (page 8), "resulting a phophoenzyme" (page 11), "should be less happened in NPC4" (page 12), "How dose NPC bind" (page 12), etc.

REVIEWERS' COMMENTS

Reviewer #2 (Remarks to the Author):

The authors went into great depths to address all my technical comments. This rectified all the unclear experimental elements that were raising concerns on the experimental rigor. The authors presented now even more solid data supporting that NPC operates in similar way to other phospholipase.

My concern on the novelty of the work is therefore partially solved but can recommend publication of this novelty they describe in the rebuttal is inserted in the main text

Response: Thank you very much for the positive comments and enthusiasm for our study. According to your new constructive suggestion, we have included the novelty-description in our revised manuscript (paragraph 2 in page 4).

Reviewer #3 (Remarks to the Author):

The authors have largely addressed all my concerns in the revised version of the manuscript. Nevertheless, I have several comments on the novel results provided by the authors.

Response: Thank you very much for your insightful comments and constructive suggestions that have helped us to largely improve our manuscript. According to your new suggestions, we further performed additional analyses and have carefully revised our manuscript. Please find our point-by-point response to your comments detailed below.

1) The authors performed all-atom molecular dynamics simulations of the NPC4 enzyme with different substrates. In Figure 4, the authors report RMSD values over the performed simulations to indicate stable models. RMSD reflects on the stability of either the enzyme or the substrate but less on the stability of the substrate-enzyme complex. The authors should show the distance between the substrate and, e.g. T158 as a function of time to assess the complex's stability.

Response: Thank you for this insightful comments. According to your suggestion, we analyzed the fluctuations of the distances between the nucleophilic atom and the phosphorus atom of substrates during the simulations. We found that these distances fluctuate slightly during the simulation time, further indicating that these docked models should be stable. We have included these analyses in over revised manuscript (paragraph 1 in page 10) and added figures as Supplementary Figure 12a-d.

2) In Figure 2b showing the NPC4 relative activity, some data points are either missing or are negative. How could be the enzyme activity negative? How does this fact affect the statistics

shown?

Response: We are sorry for missing some detailed information. In our activity assay, GIPC was used as the substrate. The GIPC was extracted from cabbage leaves and contained a small amount of GIPC-hydrolyzed product, hydroxyceramide. This contamination of GIPC substrate resulted in a detectable hydroxyceramide signal even in a blank assay (performing assay without enzyme). We used hydroxyceramide signal to assess enzyme activity. For each measurement, we subtracted the blank hydroxyceramide signal and reported the relative activity of each mutant enzyme referenced to wild-type. In Figure 2b therefore the relative enzyme activity of some mutants with low GIPC-hydrolyzing ability were negative (-1% and -5% for H79A, -6% for T158E, and -8% for D299A), due to the blank hydroxyceramide signal subtraction. To avoid negative data points affect the statistics, we defined them with zero, a data processing method used before (DOI: 10.1038/s41589-020-0499-8), and re-performed the two-side *t* test in revision. The relevant conclusion remains unchanged

Thank you for pointing this out. We have updated Figure 2b. We have also included these detailed information in our revised Methods (bottom of page 13, and page 14).

3) The authors state that Zn^{2+} ions partially reduce the NPC4 activity. However, Supplementary Fig. 8c shows a partial reduction of the activity in the presence of Mg^{2+} and not Zn^{2+} .

Response: Thank you for pointing this out. We are sorry that we have made a mistake when preparing this figure. In fact, our data show that it is the presence of Zn^{2+} ions reduce the activity, not Mg^{2+} . We denoted the data incorrectly in our previous Supplementary Fig. 8c. We have corrected this in revision.

4)It is unclear if the experiment shown in panel a of Supplementary Fig. 14 results from one or several replicas. The authors should attempt to quantify the liposome binding and the effect of truncations on binding. How would liposome association be affected by the presence of negatively charged phospholipids which are hallmarks of the plasma membrane?

Response: Results of Supplementary Fig. 14a were performed three times. According to your suggestion, we quantified the liposome binding based on the pellet and supernatant fractions resolved on SDS-PAGE gels. Results showed that these truncations had little effect on NPC4-liposome association. Thank you for this constructive suggestion. We have included these analyses in the revised Supplementary Figure 14.

Regarding to your question about the role of the presence of negatively charged phospholipids in liposome-NPC4 association, we compared the electrostatic surface of NPC4¹⁻⁴⁹⁶ with that of human PLDs (Response Fig. 1). Reported studies have showed that human PLD1 and PLD2 structures contain two polybasic regions responsible for negatively charged lipid binding/stimulating and membrane binding, respectively (DOI: 10.1038/s41589-020-0499-8, DOI: 10.1038/s41589-019-0458-4). In contrast, there is no continuous distribution of positively charged surfaces on the NPC4¹⁻⁴⁹⁶ (Response Fig. 1c), indicating its weak binding ability to membrane and / or negatively charged phospholipids. This is consistent with our previous study that NPC4 is mainly tethered to plasma membrane by a *S*-acylation modification (DOI: 10.1111/tpj.15260). Thus, we would answer your question that the presence of negatively charged phospholipids should have little effect on liposome-NPC4 association.

Furthermore, our unpublished data show that NPC4 has no detectable lipid-binding capacity in vitro (including negatively charged lipids) (Response Fig. 2).

Response Fig. 1. Electrostatic surfaces of hPLD2, hPLD1 and NPC4¹⁻⁴⁹⁶ are colored in terms of electrostatic potential, displayed in a scale from red (-5 kT/e) to blue ($+5$ kT/e).

Response Fig. 2. Lipid-NPC4 binding assay resolved on filters (left panel). Protein from the empty vector was used as a negative control. The right panel shows the positive control, of a PA-specific binding protein.

5) Newly added text contains several typos, and some sentences are hard to follow, e.g. "autoinhibited sate" (page 6), "a catalytic mechanism of substrate-hydrolysing" (page 7), "densities is" (page 8), "aside from the catalytic pocket which concerning with" (page 8), "resulting a phophoenzyme" (page 11), "should be less happened in NPC4" (page 12), "How dose NPC bind" (page 12), etc.

Response: We are sorry for these typos in our previous manuscript. We have corrected them and carefully proofread this manuscript. The revision should be most free of syntax errors.